# Knockout of Perilipin-2 in Microglia Alters Lipid Droplet Accumulation and Response to Alzheimer’s Disease Stimuli

**DOI:** 10.3390/cells14221783

**Published:** 2025-11-13

**Authors:** Isaiah O. Stephens, Lance A. Johnson

**Affiliations:** 1Department of Physiology, University of Kentucky, Lexington, KY 40508, USA; isaiah.stephens@uky.edu; 2Sanders Brown Center on Aging, University of Kentucky, Lexington, KY 40508, USA

**Keywords:** lipid droplets, microglia, Alzheimer’s disease, neuroinflammation, lipidomics, phagocytosis

## Abstract

**Highlights:**

**What are the main findings?**
Knockout of Plin2 in microglia reduces lipid droplet burden, while enhancing phagocytic clearance capacity.Transcriptomic, bioenergetic, and lipidomic analyses reveal that loss of Plin2 reprograms microglial metabolism toward reduced TAG storage and improved mitochondrial resilience.

**What are the implications of the main findings?**
Plin2 serves as a key regulator of microglial lipid droplet stability, metabolic flexibility, and immune function under Alzheimer’s-relevant stressors.Targeting Plin2 may represent a therapeutic strategy to alleviate lipid droplet-driven dysfunction and restore microglial performance in aging and neurodegeneration.

**Abstract:**

Lipid droplets (LDs) are emerging as key regulators of metabolism and inflammation, with their buildup in microglia linked to aging and neurodegeneration. Perilipin-2 (Plin2) is a ubiquitously expressed LD-associated protein that stabilizes lipid stores; in peripheral tissues, its upregulation promotes lipid retention, inflammation, and metabolic dysfunction. Yet, its role in microglia remains unclear. Using CRISPR-engineered Plin2 knockout (KO) BV2 microglia, we examined how Plin2 contributes to lipid accumulation, bioenergetics, and immune function. Compared to wild-type (WT) cells, Plin2 KO microglia showed markedly reduced LD burden under basal and oleic acid-loaded conditions. Functionally, this was linked to enhanced phagocytosis of zymosan particles, even after lipid loading, indicating improved clearance capacity. Transcriptomics revealed genotype-specific responses to amyloid-β (Aβ), especially in mitochondrial metabolism pathways. Seahorse assays confirmed a distinct bioenergetic profile in KO cells, with reduced basal respiration and glycolysis but preserved mitochondrial capacity, increased spare reserve, and a blunted glycolytic response to Aβ. Together, these findings establish Plin2 as a regulator of microglial lipid storage and metabolic state, with its loss reducing lipid buildup, enhancing phagocytosis, and altering Aβ-induced metabolic reprogramming. Targeting Plin2 may represent a strategy to reprogram microglial metabolism and function in aging and neurodegeneration.

## 1. Introduction

Lipid droplets (LDs) are dynamic intracellular organelles that serve as neutral lipid storage depots and play central roles in lipid metabolism. Structurally, LDs consist of a neutral lipid core surrounded by a phospholipid monolayer embedded with proteins that regulate lipid storage, mobilization, and organelle interactions [1,2]. They form from the endoplasmic reticulum and interact extensively with mitochondria, lysosomes, and peroxisomes, coordinating metabolic flux [1,3]. LDs respond to cellular energy demands by storing excess fatty acids or releasing them via lipolysis and lipophagy [1,4]. They buffer toxic lipid species and support immune signaling by serving as platforms for eicosanoid synthesis, including prostaglandins and leukotrienes [5,6,7]. These immunometabolic roles implicate LDs in a diverse range of diseases.

LD accumulation is a feature of many chronic inflammatory and metabolic diseases, including NAFLD, atherosclerosis, obesity, and type 2 diabetes, where it contributes to inflammation and insulin resistance [8,9,10,11]. Although LD research has historically focused on peripheral tissues, growing evidence highlights their significance in the central nervous system (CNS), particularly within glial cells such as astrocytes and microglia [12,13]. In microglia, LDs form in response to aging, metabolic stress, inflammation, or uptake of lipid-rich debris such as myelin [14]. Under these conditions, microglia can transition into a lipid-laden phenotype known as lipid droplet-accumulating microglia (LDAM), characterized by elevated LD content, impaired phagocytosis, and a pro-inflammatory transcriptional profile [12,15]. LDAMs are induced by stimuli such as lipopolysaccharide (LPS) and amyloid-beta (Aβ), which also elevate ROS and cytokine production while impairing microglial clearance functions [12,16,17]. These cells are increasingly observed with age and are enriched near amyloid plaques in Alzheimer’s disease (AD), where their abundance correlates with disease progression [18].

Plin2, a lipid droplet-associated protein, regulates lipid storage and mobilization by stabilizing droplets and limiting lipase access [19,20]. In peripheral tissues, Plin2 upregulation promotes lipid retention in settings such as fatty liver, atherosclerosis, and insulin resistance [9,21,22], while Plin2 deficiency reduces lipid burden and inflammation [23,24]. In the CNS, Plin2 is often used as an LD marker, with expression increasing with age, stroke, and AD [12,15,18], where it is prominently detected in lipid-laden microglia. However, its precise function in glial lipid biology, particularly in neurodegeneration, remains unclear, representing a critical gap with important implications for neuroinflammation and disease.

Given Plin2′s central role in lipid droplet biology and its elevated expression in aging and disease-associated microglia, we sought to define its functional contribution to microglial responses. Thus, we created a CRISPR-generated Plin2 knockout BV2 microglial model and compared it to wild-type cells under multiple AD-relevant stimulatory conditions. We demonstrate that loss of Plin2 reduces lipid burden, modulates inflammatory signaling, and restores phagocytic function in microglia. These findings identify Plin2 as a potential therapeutic target for modulating neuroinflammation in aging and neurodegeneration.

## 2. Materials and Methods

### 2.1. BV2 Cell Culture

CRISPR/Cas9-generated Plin2 knockout (Plin2 KO) BV2 microglial cells and their wild-type (WT) counterparts were generated with Ubigene (Guangzhou, China). Cells were maintained in DMEM/F-12 supplemented with 1× GlutaMAX, 10% fetal bovine serum (FBS), and 1% penicillin–streptomycin. Cultures were incubated at 37 °C in a humidified atmosphere containing 5% CO_2_ under normoxic conditions. For experiments, cells were seeded onto poly-L-lysine (PLL)-coated culture vessels (6-well plates or 8-well chamber slides). PLL was applied for 2 h at 37 °C, followed by a 1× DPBS wash and air-drying overnight.

For omics-based experiments, BV2 cells were treated with Alzheimer’s disease-relevant stimuli for 24 h prior to RNA or lipid extraction. Treatments included oleic acid (250 μM), myelin debris (15 μg/cm^2^), differentiated N2a neuron-like cells (dN2A) at a 5:1 dN2A:BV2 ratio, and amyloid-β (1.5 μM).

Oleic acid was purchased pre-conjugated to bovine serum albumin (BSA) at 100 mg/mL in DPBS (Sigma-Aldrich, St. Louis, MO, USA; Cat: 03008 Lot: SLCH8543). Stock concentration reported as 892 μg/mL, equivalent to a 3.14 mM concentration. This stock was spiked into an aliquot of growth media to a concentration of 250 μM on the day of treatment. Oleic acid-containing media were then used during media change for experimental wells/chambers. Control wells received media spiked with an equivalent volume of DPBS containing only 100 mg/mL BSA.

For the indicated conditions, BV2 cells were pre-treated with the diacylglycerol O-acyltransferase 1 (DGAT1) inhibitor A922500 (final 1.5 µM; DMSO vehicle) for 2 h, then cultured for an additional 24 h in either control medium or 250 µM BSA-conjugated oleic acid with inhibitor maintained throughout. Vehicle controls receiving DMSO matched the drug condition.

Synthetic amyloid-β (1–42) (AnaSpec (Eurofins), Fremont, CA, USA) was prepared as previously described [25]. A 1 mg stock of lyophilized amyloid-β was dissolved in 200 μL of hexafluoroisopropanol, divided into 0.2 mg aliquots, dried in a chemical fume hood, and then stored as a film at −20 °C. Immediately prior to use, amyloid-β was resuspended in Me2SO to 10 mm and water bath sonicated for 10 min. It was further prepared by diluting the amyloid-β to 25 μm with phenol red-free DMEM-F12 and incubating for 24 h at 4 °C without shaking.

N2A cells were expanded into multiple T-75 flasks with non-filter caps and grown to confluence; caps were then tightened to prevent gas exchange and induce hypoxia for 2.5 days, after which detached/apoptotic cells were harvested. Cell suspensions were pelleted, the supernatant was removed, and the pellets were resuspended and aliquoted at 1 × 10^6^ cells/mL for storage at −80 °C. For each experiment, aliquots were thawed, pelleted, the supernatant discarded, and the pellet resuspended in BV2 vehicle growth medium to the desired dose so that vehicle controls received an equal volume of the same medium. Trypan blue confirmed no viable cells before use.

Myelin debris was isolated from adult mouse brain using Percoll density gradient centrifugation as previously described [26]. Briefly, brains were homogenized in ice-cold buffer and layered onto a Percoll gradient, then centrifuged to separate myelin-rich fractions. The upper myelin-containing layer was collected, pelleted by centrifugation, and washed twice in sterile 1× DPBS by resuspension and re-pelleting. The final pellet was resuspended in 1× DPBS and homogenized by repeated passage through a fine-gauge needle to ensure uniform dispersion. Aliquots were then stored at −80 °C until the time for experimental use.

### 2.2. Mouse Model

Human APOE4 targeted-replacement (APOE4-TR) mice were crossed with 5xFAD transgenic mice to generate APOE4-TR;5xFAD (E4-5xFAD) offspring, which were aged to 12 months and used to illustrate and validate plaque associated plaque-associated Plin2 microglia in vivo. Animals were housed on a 12:12-h light/dark cycle with food and water provided ad libitum. All animal work was performed in accordance with institutional guidelines and approved by the University of Kentucky IACUC (2016-2569) on 24 September 2024.

### 2.3. Mouse Tissue Staining

Mice were deeply anesthetized and transcardially perfused with ice-cold PBS followed by 4% paraformaldehyde (PFA); brains were post-fixed overnight (4 °C), rinsed, and sectioned at 50 µm on a vibratome. Sections were blocked/permeabilized for 1 h at room temperature in 10% donkey serum in PBS + 0.2% Triton X-100, then incubated overnight (4 °C) in 3% donkey serum in PBS-T with goat anti-Iba1 (Novus NB100-1028, 1:500) and rabbit anti-PLIN2/ADRP (Novus Biologicals, Centennial, CO, USA, NB110-40877, 1:500). After three 10-min PBS-T washes, sections were incubated 2 h at room temperature (protected from light) with donkey anti-goat Alexa Fluor 555 (Thermo Fisher Scientific, Waltham, MA, USA; A32816, 1:200) and donkey anti-rabbit Alexa Fluor 488 (Thermo Fisher A32790, 1:200), rinsed in PBS, counterstained with AmyloGlo (Avantor 76264-656, 1:100 in PBS, ~10 min), and mounted in EverBrite TrueBlack Hardset with NucSpot 640 (Biotium, Fremont, CA, USA; 23019). Representative confocal images (Nikon Instruments Inc., Melville, NY, USA; AXR; 40× Water immersion objective for whole brain image and 60× oil immersion objective for zoomed image) were acquired to illustrate lipid-laden microglia surrounding amyloid beta plaques.

### 2.4. Western Blot

WT and Plin2 KO BV2 cells were treated with 250 µM oleic acid for 24 h, washed twice with ice-cold PBS, and lysed on ice in RIPA buffer (50 mM Tris-HCl pH 7.5, 150 mM NaCl, 1% NP-40, 0.5% sodium deoxycholate, and 0.1% SDS) supplemented with protease/phosphatase inhibitors; lysates were clarified at 16,000× *g* for 15 min at 4 °C and protein was quantified by BCA. Equal protein (15 µg) was denatured in 4× Laemmli buffer with 5% β-mercaptoethanol (95 °C, 5 min), separated on 4–20% Criterion TGX precast gels (Bio-Rad Laboratories, Hercules, CA, USA; 5671094) at 180 V for 50–65 min, and transferred to nitrocellulose using Trans-Blot Turbo midi transfer packs (Bio-Rad Laboratories, Hercules, CA, USA; 17041590) with the manufacturer’s rapid program (~7 min). Membranes were blocked in casein blocking buffer (Thermo Fisher Scientific, Waltham, MA, USA; PI37528) for 1 h at room temperature and incubated overnight at 4 °C with rabbit anti-PLIN2/ADRP (Novus Biologicals, Centennial, CO, USA; NB110-40877, 1:800) diluted in blocking buffer; after three 5-min washes in PBS-T, membranes were incubated for 1 h at room temperature (protected from light) with StarBright Blue 700 goat anti-rabbit IgG (Bio-Rad, 12004161; 1:1000), washed, and imaged on a ChemiDoc system using the near-IR 700-nm channel with exposures set within the linear range. A single representative blot is shown to illustrate the complete absence of Plin2 signal in Plin2 KO cells.

### 2.5. Lipid Droplet Staining

For lipid droplet (LD) visualization, BV2 cells were treated with 250 μM oleic acid (OA) or 1× DPBS as a control for 24 h in glass chamber slides. After treatment, cells were washed with 1× DPBS and incubated in OptiMEM containing 0.15 μM Lipi-Green (Dojindo Molecular Technologies, Rockville, MD, USA) for 30 min at 37 °C. Cells were then fixed in 4% paraformaldehyde (PFA) for 15 min, washed three times with 1× DPBS, and mounted using Vectashield HardSet Antifade Mounting Medium with DAPI. Slides were cured overnight and imaged using a Nikon (Melville, NY, USA) AXR confocal microscope (University of Kentucky Light Microscopy Core).

### 2.6. LD Time Course Experiment

BV2 microglial cells were seeded on glass chamber slides in standard growth medium and allowed to adhere overnight at 37 °C/5% CO_2_. The following morning, cells were swapped to media with either 250 µM oleic acid or 1.5 µM amyloid-β in growth medium. Slides were removed at 6, 18, and 24 h post-treatment and washed once with 1× DPBS before incubation in OptiMEM containing 0.15 µM Lipi-Green for 30 min at 37 °C. Cells were then fixed in 4% paraformaldehyde for 15 min, washed three times with 1× DPBS, and mounted in Vectashield HardSet with DAPI. Immediately after the 24-h synthesis time point, remaining slides were washed and switched to serum-free media to promote LD degradation; these were processed at +2, +6, +18, and +24 h after serum removal using the identical Lipi-Green staining, fixation, and mounting steps. All images were acquired on a Nikon (Melville, NY, USA) AXR confocal microscope (University of Kentucky Light Microscopy Core).

### 2.7. Phagocytosis Assay

Following the 24 h treatment, with 250 μM OA, 1.5 μM amyloid-β, or vehicle-treated cells in 8-chamber slides were incubated with either 15 μL of 1× DPBS (vehicle) or 15 μL of pHrodo™ Red Zymosan Bioparticles (Abcam, Cambridge, UK) per well. After a 2 h incubation at 37 °C, cells were washed and fixed with 4% PFA for 15 min, then mounted using Vectashield hardset mounting media with DAPhI and allowed to cure overnight, hidden from light. Slides were then prepared and imaged on the Nikon W1 spinning disk confocal microscope using a 40× water immersion objective (University of Kentucky Light Microscopy Core).

### 2.8. Image Analysis

All confocal images were uploaded as ND2 files and analyzed in HALO (Indica Labs, Albuquerque, NM, USA) using the Co-localization FL (v3.5) algorithm. For each experiment, a single contrast threshold was set per relevant channel to exclude background/autofluorescence; after parameter lock, the same contrast threshold and the module’s minimum/maximum object-size settings were applied identically to all images, with the analyst blinded to group. Lipi-Green was quantified as both the total Lipi-Green-positive area and the number of discrete Lipi-Green objects (particles) using the fixed contrast and size gates. pHrodo was quantified as the total pHrodo-positive area. All readouts were normalized to the number of DAPI-positive nuclei (per-cell values).

### 2.9. Lipidomics

BV2 cells were plated on poly-l-lysine-coated 6-well plates and grown to ~75% confluence at 37 °C, 5% CO_2_. On the day of extraction, the plates were removed from the incubator and placed in the biosafety cabinet. Medium was aspirated, wells were rinsed once with 500 µL ice-cold 1× DPBS and aspirated again, then 500 µL of pre-chilled extraction solvent (50:50 methanol–butanol, 10 mM ammonium formate) was added to each well. Plates were transferred to –80 °C for 15 min, then cells were scraped from the first 500 µL and transferred to 1.5 mL tubes. A second 500 µL solvent wash was used to rinse each well; that wash was pooled into the same tube (≈1 mL total). Tubes were vortexed briefly and stored at –80 °C until the day of analysis. On analysis day, samples were thawed on ice, centrifuged at 14,000× *g* for 10 min at 4 °C to pellet debris, and the clear supernatant was transferred to LC-MS vials. Lipidomic data were acquired on an Agilent QQQ LC-MS system (Agilent, Santa Clara, CA, USA).

### 2.10. Transcriptomics

Total RNA was isolated from BV2 cells using the Qiagen RNeasy Mini Kit (Qiagen, Hilden, Germany). RNA quality was assessed, and samples with RIN ≥ 8 were submitted to Novogene for poly(A)-enriched paired-end (2 × 150 bp) Illumina sequencing. Reads were aligned to the mouse reference genome (GRCm38/mm10), and gene counts were obtained. Differential gene expression levels were analyzed in R using the DESeq2 package. Genes with FDR-adjusted *p*-values < 0.05 and fold change ≥ 1.5 were considered differentially expressed. Principal component analysis (PCA), hierarchical clustering, and Gene Ontology (GO) enrichment analyses were performed using prcomp, pheatmap, and clusterProfiler packages, respectively. All data can be found in Appendix A.

### 2.11. Co-Expression Network Construction

BV2 bulk RNA-seq counts and metadata were loaded into R. Genes with CPM > 1 in ≥25% of samples, zero-variance genes, and Rps/Rpl transcripts were removed. Raw counts were variance-stabilized with DESeq2′s vst (dds, blind = TRUE), outliers were removed by hierarchical clustering, and the cleaned matrix was used for WGCNA module detection. A signed co-expression network was built in WGCNA (v1.73) using soft-threshold power 6 (scale-free R^2^ ≥ 0.8), blockwiseModules (minModuleSize = 30, deepSplit = 3, mergeCutHeight = 0.20), and TOMType = “signed.” Module eigengenes were correlated with Genotype_Treatment groups (Pearson’s r; *p*-values from corPvalueStudent), and intramodular connectivity (kME) defined hub genes. Module gene sets were tested for GO Biological Process enrichment with clusterProfiler (BH-adj *p* < 0.05, q < 0.05, minGSSize = 5). Adjacency and node-attribute tables (module color and kME) for each module and for hub-gene subsets (top 10 ranked by positive kME) were exported via exportNetworkToCytoscape (WGCNA v1.73) for visualization.

### 2.12. Seahorse Extracellular Flux Assay

BV2 cells (WT and Plin2 KO) were seeded at 8 × 10^3^ cells/well in Agilent XF96 plates and allowed to adhere overnight at 37 °C/5% CO_2_. Cells were then treated ±1.5 µM amyloid-β for 24 h, washed twice with Seahorse XF DMEM assay medium (10 mM glucose, 2 mM l-glutamine, 1 mM sodium pyruvate, pH 7.4), and equilibrated at 37 °C (no CO_2_) for 45 min. OCR (pmol/min) and ECAR (mpH/min) were measured on a XFe96 Analyzer using the XF Cell Mito Stress Test (1 µM oligomycin, 2 µM FCCP, 0.5 µM rotenone/antimycin A), with three 3 min mix–2 min wait–3 min measure cycles per phase (cycles 1–3 basal, 4–6 ATP-linked, 7–9 maximal, 10–12 non-mitochondrial). We calculated basal respiration, ATP production (basal–oligomycin), proton leak (oligo–non-mito), maximal respiration (FCCP–non-mito), spare respiratory capacity [(maximal–basal)/basal × 100], and coupling efficiency (ATP production/basal × 100). Each condition was run in up to 12 wells (injector-failure wells excluded), and OCR and ECAR values were normalized to DAPI-stained nuclear counts. Source values are provided in Appendix A.

### 2.13. Statistics

Unless otherwise noted, all tests were two-sided with α = 0.05. Data are presented as mean ± SEM, and *n* denotes biological replicates as specified in figure captions. For imaging-based assays (Lipi-Green lipid droplets and pHrodo uptake), group differences were analyzed by two-way ANOVA with factors Genotype (WT, Plin2 KO) and Treatment (as indicated), followed by Šídák’s multiplicity-adjusted post hoc comparisons. For each metric (LD area per cell, count per cell, average area), values were log2-transformed, centered to each run’s WT mean, then back-transformed. This aligns runs while preserving group differences. For time course experiments (OA or Aβ), we fit two-way ANOVA models with Genotype and Time within each treatment; when a Genotype × Time interaction was significant, simple effects were tested with Šídák adjustment. Assumptions (approximate normality of residuals and homoscedasticity) were checked on model residuals; variance-stabilizing transforms were applied when appropriate.

Seahorse XF analyses (OCR/ECAR metrics) were evaluated by two-way ANOVA with factors Genotype and Aβ exposure and Šídák post hoc comparisons; values were normalized to nuclei counts prior to analysis. Well-level source data and summary statistics are provided in Appendix A.

Bulk RNA-seq differential expression was assessed as per-treatment KO vs. WT contrasts. For each gene, we report log2 fold change (KO−WT), *p*-value, and Benjamini–Hochberg false discovery rate (BH-FDR); the primary discovery threshold was FDR < 0.05, with effect-size emphasis at |log2FC| ≥ 0.585 (~1.5×) where noted. Complete gene-level results are in Appendix A, with raw/normalized counts in the accompanying sheet.

Weighted gene co-expression network analysis (WGCNA) module–trait associations were quantified by Pearson’s *r* with BH-adjusted *p*-values. Hub genes were defined by the highest intramodular connectivity (kME). Module membership tables and enrichment outputs are provided in Appendix A.

Lipidomics (species and subclass levels). Species were analyzed with per-feature linear models estimating KO−WT log2FC within each treatment, reporting t, df, *p*, and Benjamini–Hochberg (BH) FDR (primary threshold FDR < 0.05). Subclass totals were tested by two-way ANOVA (Genotype, Treatment, Genotype × Treatment) with BH FDR across subclasses per term; KO−WT pairwise contrasts were computed within each treatment and BH-adjusted across subclasses within that treatment. Full per-species, subclass pairwise, and subclass ANOVA outputs are provided in Appendix A.

## 3. Results

### 3.1. Plin2 KO Lowers LD Accumulation and Size in BV2 Cells After Oleic Acid Stimulation

To establish the role of Plin2 in microglial LD biology, we first examined its expression in vivo. Similar to previous reports [16,18], immunofluorescence staining in 5xFAD mice expressing human APOE4 (the major AD-risk isoform) revealed Plin2-positive microglia clustered around amyloid plaques, where they contained abundant LDs (Figure 1a). To directly test the functional contribution of Plin2, we first confirmed the loss of Plin2 in our KO microglia, with loss of the 48 kDa Plin2 protein confirmed by Western blotting (Figure 1b). We then employed an oleic acid (OA)–loading paradigm to assess LD formation, as outlined in the experimental workflow schematic (Figure 1c). Under basal conditions, both WT and Plin2 KO cells exhibited few LDs. Following 24 h of OA treatment, WT cells displayed a robust increase in total LD area, whereas this accumulation was significantly attenuated in Plin2 KO cells (Figure 1d,e). OA treatment increased the LD count per cell in both genotypes (Figure 1f), and the average size per cell was only increased in the WT (Figure 1g). The number of LDs per cell and the average droplet size were both significantly reduced in KO cells compared to WT (Figure 1f,g). Together, these findings demonstrate that Plin2 is required for efficient expansion of lipid droplet size, number, and accumulation in microglia following an exogenous lipid challenge.

### 3.2. Phagocytosis of Zymosan Particles Is Enhanced in Plin2 KO Cells

Lipid-laden phagocytes, including LDAMs, have been shown to be deficient in their ability to clear various targets [12]. To determine if the loss of Plin2 and lower accumulation of LDs influences phagocytic capacity, we tested the ability of Plin2 KO cells to clear zymosan particles under basal and OA-loaded conditions. Zymosan, a yeast-derived particle that robustly engages pattern recognition receptors, is a standard phagocytosis assay and provides a reliable readout of microglial clearance function in this context [12,27,28]. Under control conditions, Plin2 KO cells internalized significantly more zymosan than WT (Figure 2a,b). Treatment with 250 µM oleic acid for 24 h did not affect WT uptake (WT + OA vs. WT control), yet further increased phagocytosis in KO cells. Even after lipid loading, KO cells exhibited substantially greater uptake than WT + OA, demonstrating that Plin2 constrains phagocytic capacity in BV2 microglia under both normal and lipid-loaded conditions. Together, these results underscore the impact of lipid accumulation on phagocytic function and suggest that reducing Plin2 may augment microglial clearance capacity.

### 3.3. Bulk RNA-Sequencing Shows Plin2 Alters BV2 Response Profiles to AD-Relevant Stimuli

To investigate how loss of Plin2 reprograms microglial transcriptional programs, we performed bulk RNA-seq under basal and stimulated conditions and complemented this with Weighted Gene Co-expression Network Analysis (WGCNA) to resolve coordinated gene networks. Bulk RNA-seq revealed clear transcriptional divergence between Plin2 KO and WT microglia, with the extent of separation varying by treatment (Figure 3a,b). At large, the KO cells showed upregulation of innate immune, chemotaxis, and migration pathways, while sterol and cholesterol biosynthesis and proliferative programs were downregulated (Figure 3c,f–h). Importantly, this immune versus lipid metabolic split was evident not only at baseline but also under treatment with myelin and apoptotic neurons. By contrast, Aβ exposure elicited a distinct transcriptional response: KO cells upregulated mitochondrial and oxidative phosphorylation pathways, while small GTPase-mediated signaling and cytoskeletal regulation were downregulated (Figure 3d,e).

WGCNA further resolved these networks, identifying KO-associated modules (cyan, pink, tan, black) that were consistently upregulated across all treatment conditions, and capturing programs related to Fc receptor signaling, innate immune activation, cytoskeletal remodeling, and cell cycle regulation (Figure 4c,d). Notably, the cyan module contained Fcer1g, a central adaptor for Fc receptor signaling, while the tan module included DNA replication and cell cycle regulators such as Mcm3 and Cdc6 (Figure 4d). Conversely, modules that were consistently downregulated in KO (purple, red, green-yellow, magenta) contained pathways linked to sterol and cholesterol biosynthesis, TNF-driven cytokine signaling, and adaptive immune activation (Figure 4c). Key hub genes within these downregulated modules included canonical lipid regulators (Hmgcr, Hmgcs1) and immune mediators (Clec7a, Spp1) (Figure 4e). Additionally, a single module (green) captured a mitochondrial–endosomal signature that was enriched in KO only under Aβ and dN2A but not under Control, OA, or Myelin (Appendix A), paralleling the amyloid-specific GO differences between genotypes in bulk RNA-seq. Prompted by these Aβ-specific differences and the Fc receptor-enriched module, we performed the zymosan phagocytosis assay after pre-treatment with Aβ for 24 h and found that the Plin2 KO cells still phagocytose/ingest more particles than the WT counterpart (Appendix A).

Taken together, bulk RNA-seq and WGCNA demonstrate that Plin2 shapes both metabolic and immune programs in microglia. Across multiple stimuli, KO cells consistently downregulated sterol and lipid biosynthetic pathways while upregulating immune and Fc receptor-associated networks, with the strongest divergence observed under Aβ challenge. These transcriptomic differences provide a mechanistic framework for the altered lipid handling and phagocytic phenotypes observed in Plin2 KO microglia.

### 3.4. Plin2 Regulates Lipid Droplet Dynamics in Response to Oleic Acid and Amyloid-β

As our transcriptomic analyses indicated that Aβ elicited the strongest genotype-specific transcriptional response in Plin2 KO versus WT microglia, we next examined whether these differences extended to LD dynamics. We performed a two-phase time course in our BV2 cells treated with 250 µM OA or 1.5 µM Aβ, measuring LD synthesis at 6, 18, and 24 h, followed by a serum-free chase at +2, +6, +18, and +24 h to monitor degradation.

Under OA, Plin2 KO cells showed significantly decreased LD accumulation compared to WT at 24 h and +2 h (Figure 5c) (Appendix A). After serum removal, both genotypes showed progressive LD clearance, returning to baseline by +24 h; however, KO cells maintained consistently lower LD levels throughout the time course. With Aβ, KO cells showed no significant LD accumulation during the first 24 h, whereas WT displayed a marked increase, detectable at 6, peaking at 18, and still elevated at 24 h (Figure 5d). Together, these findings show that loss of Plin2 limits LD accumulation in response to both OA and Aβ, underscoring its role in regulating microglial lipid storage in response to diverse stimuli.

### 3.5. Mitochondrial Respiration in BV2 Microglia Under Basal and Amyloid-β Challenge

Given the mitochondrial gene expression differences and unique Aβ response, we next assessed bioenergetics in Plin2 KO microglia. KO cells exhibited significantly reduced basal respiration under both control and Aβ-treated conditions (Figure 6a,c). Despite this lower baseline, KO cells maintained the ability to reach similar or higher levels of maximal respiration (Figure 6d), resulting in a greater spare respiratory capacity. Proton leak was consistently lower at baseline and after Aβ exposure (Figure 6e), reflecting tighter coupling efficiency. For glycolysis, WT cells showed a trend toward higher basal ECAR, and upon Aβ exposure exhibited a significant increase, whereas this elevation in glycolysis was blunted in KO cells (Figure 6b,f). Integration of ECAR and OCR data (Figure 6g) shows that Plin2 loss establishes a lower energetic set-point—both basal OCR and ECAR are reduced—while preserving respiratory capacity. This profile potentially reflects a more efficient baseline with an enhanced ability to meet sudden energetic demands. In contrast, WT microglia allocate toward higher baseline expenditure and mobilize glycolysis in response to Aβ. Together, these findings suggest that Plin2 functions as a calibrator of microglial energy budgeting: promoting higher baseline activity and glycolytic responsiveness under Aβ treatment.

### 3.6. Plin2 KO Significantly Alters the Microglia Lipidome

Because Plin2 regulates neutral lipid storage, we next performed targeted lipidomics to define how its loss reprograms the BV2 lipidome. Across conditions, including control, oleic acid, and dN2A, KO cells consistently showed lower levels of triacylglycerols (TAGs) and diacylglycerols (DAGs), while also having relative enrichment of cholesteryl esters (CE) (Figure 7b–g). Under myelin treatment, however, CE levels did not differ between genotypes, though KO cells continued to display reduced TAG/DAG abundance (Figure 7g). In our Aβ-treated samples, we see a similar trend as before, where the KO cells have significantly lower TAG/DAG abundance with an increase in abundance of CE. Previous studies have shown that LD composition is primarily composed of TAG (~90%) [29]. Thus, the significantly lower abundance of TAG/DAG seen in the KO samples is likely the driving force for the lower LD accumulation. This was confirmed by treating cells with an inhibitor for DGAT1, which led to no significant difference in LD accumulation after being treated with OA (Appendix A). This result also confirms that the increase in CE found in the KO cells does not contribute significantly to the LD area. Overall, these data support the conclusion that Plin2 promotes TAG-rich droplet storage in microglia, while its loss reduces neutral lipid accumulation. Additionally, loss of Plin2 modulates cholesterol handling in vitro in microglia under these specific conditions tested.

## 4. Discussion

Microglia play a central role in maintaining brain homeostasis through surveillance, clearance, and immune regulation. A growing body of work demonstrates that LD accumulation fundamentally alters these functions [12,16,30,31]. LD-rich microglia emerge in aging and AD, where they exhibit impaired phagocytosis, heightened oxidative stress, and pro-inflammatory bias [12,15,30,32]. Despite this recognition, the molecular scaffolds that stabilize LDs remain poorly understood in microglia. Our study identifies the LD coat protein Plin2 as a key determinant of this phenotype, showing that its loss limits droplet expansion and reprograms microglial function under diverse metabolic and disease-relevant challenges. Specifically, we show that loss of Plin2 lowers LD accumulation through reduction of TAG abundance, improves phagocytosis capacity, and yields a unique transcriptional and metabolic profile in response to Aβ in microglia.

Work in macrophages and hepatocytes has established Plin2 as a structural protein that shields lipid droplets from lipolysis [23,24,33,34], thereby promoting triglyceride retention. Our results support this paradigm in microglia. Under oleic acid exposure, WT microglia primarily expanded droplets by increasing size, whereas Plin2 KO cells maintained smaller droplets. This pattern highlights a critical role of Plin2 in stabilizing the growth of LDs. Time course analyses reinforced this conclusion: while WT cells progressively accumulated LDs with oleic acid and exhibited a transient burst under Aβ treatment, KO cells remained comparatively resistant to expansion. Together, these findings support a model in which Plin2 acts as a molecular brake on LD turnover, sustaining accumulation in conditions of lipid excess or amyloid stress.

Lipid-laden phagocytes are consistently characterized by reduced clearance capacity, and defective phagocytosis is a hallmark of AD microglia [12,30]. Our results demonstrate that Plin2 KO enhances uptake of zymosan particles under both basal, lipid-loaded conditions, and after treatment with Aβ. Zymosan engages CLEC7A/TLR2 and Fc receptors [35], providing a broad assay of innate uptake capacity. The improved clearance in KO cells mirrors transcriptomic signatures enriched for Fc receptor signaling, innate immune pathways, and actin remodeling, all of which support recognition and internalization. Importantly, inflammatory effectors such as CCL chemokines and matrix metalloproteinases were lower in KO cells, suggesting that loss of Plin2 decouples lipid droplet accumulation from pro-inflammatory output. In addition, PLIN2 binds arachidonic acid and decorates lipid droplets that serve as COX/LOX eicosanoid-synthesis sites, supporting the idea that PLIN2 can influence lipid-mediator output that modulates phagocytosis [36,37,38]. These observations align with the broader literature in which LD-rich microglia in AD exhibit compromised clearance and inflammatory bias but add a novel mechanistic insight: Plin2 may contribute directly to this dysfunction by stabilizing the droplets that restrain immune adaptability.

Bulk RNA-seq revealed that Plin2 knockout consistently drove down sterol and cholesterol biosynthetic pathways while upregulating programs for chemotaxis, migration, RNA/ribonucleoprotein metabolism, and mitochondrial organization. Importantly, under amyloid challenge, KO cells mounted a distinctive transcriptional response characterized by enrichment of mitochondrial respiratory chain assembly, oxidative phosphorylation, and RNA splicing. This suggests that loss of Plin2 does more than blunt lipid storage; it also enables a transcriptional program that preserves mitochondrial integrity and RNA metabolism in the face of Aβ stress. In the context of AD, where LD-rich microglia are marked by oxidative stress, defective energy production, and impaired clearance [5,12,39], the KO amyloid signature represents a potentially protective adaptation that contrasts sharply with the lipid-burdened state.

WGCNA reinforced this shift at the network level. Down in KO were modules for sterol/cholesterol biosynthesis and TNF superfamily cytokine production, features that mirror the maladaptive, pro-inflammatory, lipid-burdened signature in aging and AD. By contrast, up in KO modules included Fcγ receptor signaling, innate immune pathways, and Arp2/3-mediated actin remodeling, networks central to receptor-driven phagocytosis, cytoskeletal remodeling, and debris clearance. These adaptive modules parallel those described in microglia that retain phagocytic competence under demyelination and amyloid challenge [14,16]. Additionally, a single module (green) captured a mitochondrial–endosomal program that was enriched in KO only under Aβ and dN2A and not under Control, OA, or Myelin (Appendix A); this selective induction mirrors the amyloid-contingent mitochondrial GO signal seen in bulk RNA-seq (Appendix A). Together, these data suggest that Plin2 loss gates an oxidative/endo-lysosomal program specifically under amyloid or neuron-derived stress. This selective mitochondrial–endosomal program prompted us to test whether these network shifts manifest at the bioenergetic level.

Consistent with this network signal, WT cells mounted strong glycolytic responses to Aβ, consistent with literature showing that inflammatory activation depends on aerobic glycolysis via the mTOR–HIF1α axis [40]. Yet this shift, while acutely supportive of cytokine production, is maladaptive with chronic exposure, ultimately driving the collapse of both glycolysis and OXPHOS [41]. Plin2 KO cells blunted this glycolytic surge while maintaining maximal respiration and spare capacity. This “low-glycolysis, high-reserve” profile could reflect a form of metabolic resilience, where mitochondrial flexibility is preserved rather than sacrificed for short-term inflammatory gain. Such a state may enable sustained phagocytic surveillance under prolonged amyloid stress, a functional contrast to the exhausted lipid-laden phenotype. Further work is needed to fully elucidate these effects under both acute and chronic inflammatory responses.

Targeted lipidomics further showed the significance of removing Plin2. Across control, oleic acid, Aβ, and dN2A cells, WT were enriched for TG and DG, whereas Plin2 consistently showed higher levels of CEs. The simplest interpretation is that Plin2 loss shifts neutral lipid routing away from TG storage and promotes its utilization. This is consistent with much of the previous work showing that loss of Plin2 enhances the turnover of LDs largely through lipolysis [24,42], and likely through the process of lipophagy [19]. Interestingly, these results match a previous report where fasted Plin2 KO mice had substantial reductions in TG species with modest increases in CEs [43]. However, when mice from the same study were given a high-fat diet, Plin2 KO still resulted in reduced TGs, where CEs were unchanged. Another study has specifically addressed the effect of removing Plin2 on cholesterol toxicity in bone marrow-derived macrophages and found it to be well tolerated, like our current study [44]. This evidence suggests the blunted LD accumulation seen in response to Plin2 ablation is largely driven by a reduction in acylglycerols, though increased efflux of cholesterol from Plin2 KO macrophages has also been reported [24,45]. In support of this, we show that pharmacologic DGAT1 inhibition reduced OA-induced LD area and size, with a larger effect in WT than KO. Suggesting the differences in LD area between the two genotypes are in fact driven by TG accumulation, and that the modest increase of CE in the KO samples does not lead to increased LD accumulation. With this information, the CE increase fits a feedback model in which efficient cholesterol trafficking and export reduce the need for de novo synthesis, with CE serving as a short-lived reservoir for subsequent hydrolysis and efflux [24,45]. This aligns with the observed downregulation of sterol-synthesis pathways seen in our bulk sequencing analysis. However, we also want to emphasize that these results are in the context of our specific in vitro conditions. Taken together, this suggests that Plin2 also plays a significant role in cholesterol handling in microglia, which is of particular interest in neurodegenerative disorders. A full understanding of the role of microglial Plin2 in these processes will require further investigation in vivo.

We believe our data support the growing trend of targeting LDs for their role in AD, and that interventions which reduce maladaptive droplet accumulation can improve microglial performance [24,30]. Lipid-laden microglia are a hallmark of the aged brain and are observed near plaques and in tauopathy, where droplet-laden microglia show impaired clearance and pro-inflammatory signatures [12,15,18,32]. This phenotype is further amplified by genetic risk: the strongest common risk factor for late-onset AD, APOE4, is linked to lipid droplet-rich microglial states [46]. Our data suggest that loss of Plin2 acts as a lever on this axis, pushing microglia away from chronic lipid storage and toward a more clearance-competent, metabolically resilient state. Thus, we view Plin2 as a novel therapeutic target for lowering lipid-laden microglia in AD, a unique approach that does not inhibit droplet formation but instead promotes rapid turnover and utilization.

Our study has limitations, including several inherent in the BV2 model. As a transformed cell line, it cannot capture the full heterogeneity, cell-cell interactions, and chronic exposures that shape primary murine or human microglia in vivo [47,48,49]. Future studies should extend these findings to human systems and in vivo contexts. Priority directions include: (i) validation in primary and iPSC-derived microglia, including under chronic lipid stress paradigms (e.g., prolonged oleate or mixed Aβ/myelin exposures); (ii) direct quantification of cholesterol efflux kinetics using defined acceptors (apoA-I/HDL) and labeled cholesterol tracers; (iii) measurement of lipophagy flux in the Plin2-deficient background to define links between neutral-lipid turnover and autophagy; and (iv) testing central-nervous-system–restricted Plin2 modulation in disease models amyloid- and/or tau-driven pathologies, to determine translational relevance.

## 5. Conclusions

This work establishes Plin2 as a central regulator of microglial lipid droplet biology, demonstrating that its loss reshapes transcriptional, metabolic, and functional programs. Removing Plin2 prevents microglia from becoming lipid-burdened, a state that in aging and AD is linked to impaired phagocytosis, chronic inflammation, and metabolic collapse. Instead, Plin2 knockout drives microglia toward a putatively adaptive profile. Therapeutically, modulating Plin2 or droplet turnover could offer a strategy to restore microglial performance in neurodegeneration. This approach is unique in that it lowers the lipid burden while still retaining the ability to generate LDs for lipid buffering and regulation. Together, our data define a novel mechanism by which droplet stability controls microglial fate and provide a foundation for targeting lipid-burdened microglia in disease.

## Figures and Tables

**Figure 1 cells-14-01783-f001:**
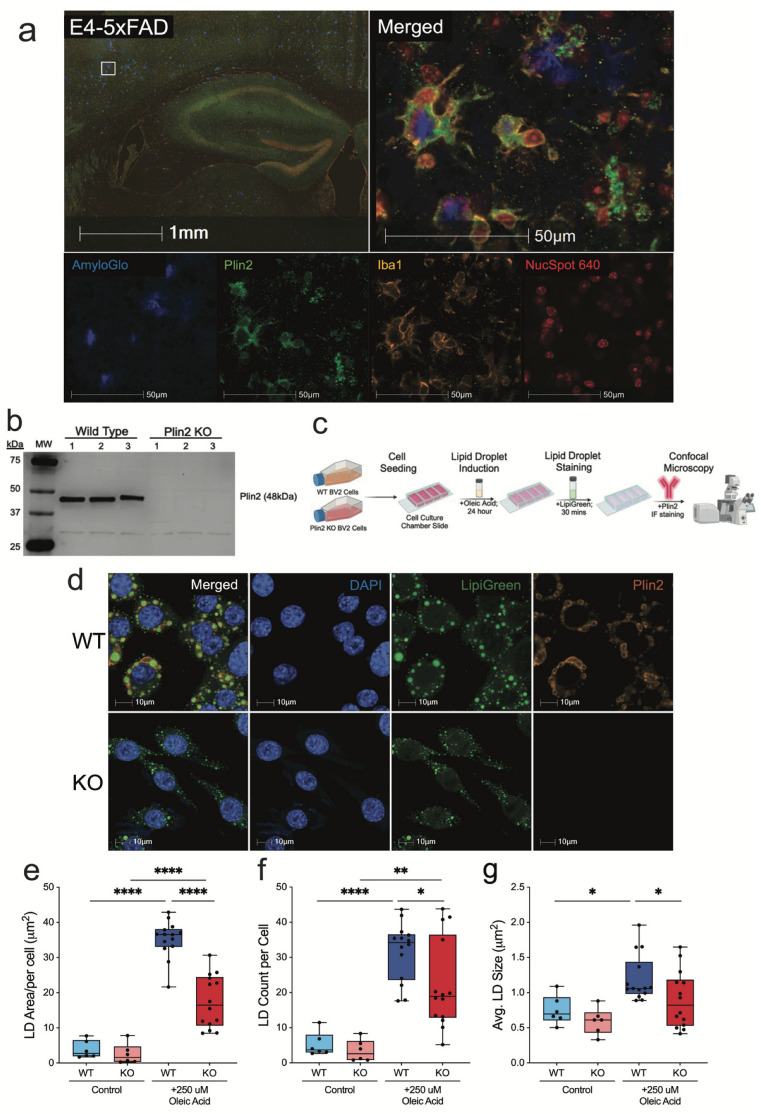
Plin2 knockout reduces oleic acid–induced lipid droplet accumulation in BV2 microglia. (**a**) Immunofluorescent staining of E4-5xFAD (APOE4-TR E4/E4; 5xFAD) brain sections showing Plin2-positive (green) microglia (Iba1, yellow) enriched around amyloid plaques (AmyloGlo, blue). Nuclei marked with NucSpot 640 (red). Inset highlights plaque-associated Plin2+ microglia. (**b**) Western blot validation of Plin2 knockout (KO) in BV2 cells using CRISPR–Cas9, confirming loss of the ~48 kDa Plin2 protein. (**c**) Experimental workflow for oleic acid (OA) loading, lipid droplet staining, and confocal imaging. (**d**) Representative confocal images of WT and Plin2 KO BV2 cells under OA-loaded conditions (Lipi-Green, green; Plin2, orange; DAPI, blue). (**e**–**g**) Quantification of lipid droplet metrics in WT and KO cells: (**e**) total lipid droplet area per cell; (**f**) number of lipid droplets per cell; (**g**) average lipid droplet size. Stats (**e**–**g**): two-way ANOVA (Genotype × Treatment) with Šídák multiplicity-adjusted post hoc tests; two-sided α = 0.05; mean ± SEM; Control *n* = 6 wells; Oleic Acid *n* = 15 wells. Exact adjusted *p*-values and full ANOVA tables are provided in Appendix A. For all statistics, * *p* < 0.05; ** *p* < 0.01; **** *p* < 0.0001.

**Figure 2 cells-14-01783-f002:**
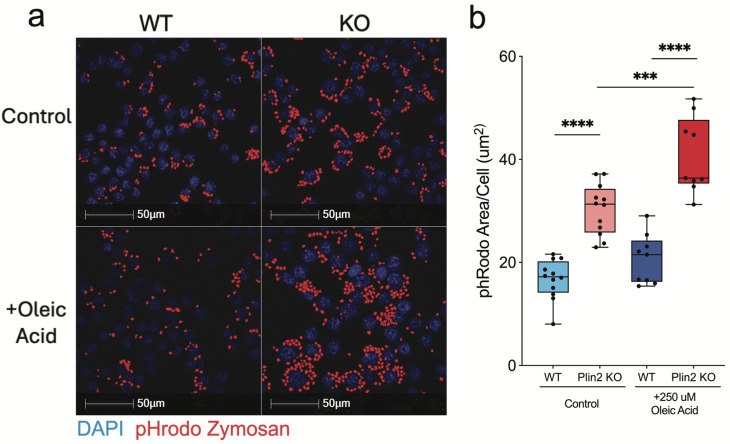
Oleic acid–dependent enhancement of zymosan uptake in Plin2-deficient microglia. BV2 cells (WT and Plin2 KO) were exposed for 24 h to vehicle or 250 µM oleic acid (OA), then challenged with pHrodo Red Zymosan Bioparticles for 2 h. (**a**) Representative confocal images of internalized bioparticles (red) and DAPI-stained nuclei (blue). (**b**) Mean pHrodo-positive area per cell quantified from *n* = 12 wells per condition. Analysis was performed on well means. Stats: two-way ANOVA (Genotype × Treatment) with Šídák multiplicity-adjusted post hoc tests; two-sided α = 0.05; mean ± SEM; Exact adjusted *p*-values and full ANOVA tables are provided in Appendix A. For all statistics, *** *p* < 0.001; **** *p* < 0.0001.

**Figure 3 cells-14-01783-f003:**
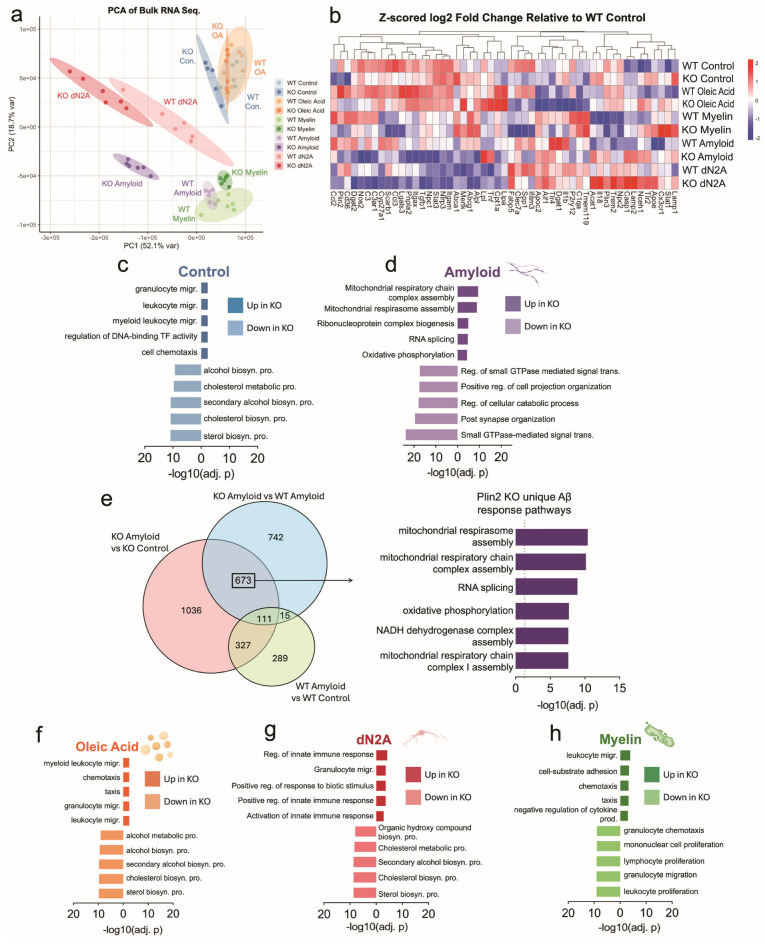
Plin2 deletion reprograms microglial transcriptomes across stimuli. (**a**) Principal component analysis (PCA) of bulk RNA-seq from WT and Plin2 KO BV2 cells under control, amyloid beta (Aβ), oleic acid (OA), dN2A, and myelin treatments. (**b**) Heat map of microglial genes and representative differentially expressed genes (DEGs), shown as Z-scored log2 fold change relative to WT control (scale at right). (**c**,**d**,**f**–**h**) Pathway enrichment analyses for DEGs in Control (**c**), Aβ (**d**), OA (**f**), dN2A (**g**), and Myelin (**h**). Bars show the top enriched Gene Ontology terms for genes up in KO (right bars) and down in KO (left bars), ranked by −log10(adjusted *p* value). Across conditions, sterol and broader lipid-biosynthetic programs were KO-depleted, whereas innate immune, cytoskeletal, vesicle/lysosome, and—under Aβ—mitochondrial respiration/respirasome pathways were KO-enriched. (**e**) Venn diagram of DEGs under Aβ exposure across three contrasts: KO Aβ vs. KO Control, WT Aβ vs. WT Control, and KO Aβ vs. WT Aβ. Stats: per-treatment KO vs. WT contrasts with Benjamini–Hochberg false discovery rate (BH FDR) control; primary threshold FDR < 0.05. Functional enrichment used GO over-representation with BH-adjusted *p*-values (FDR control). Complete gene-level results and enrichment tables are provided in Appendix A.

**Figure 4 cells-14-01783-f004:**
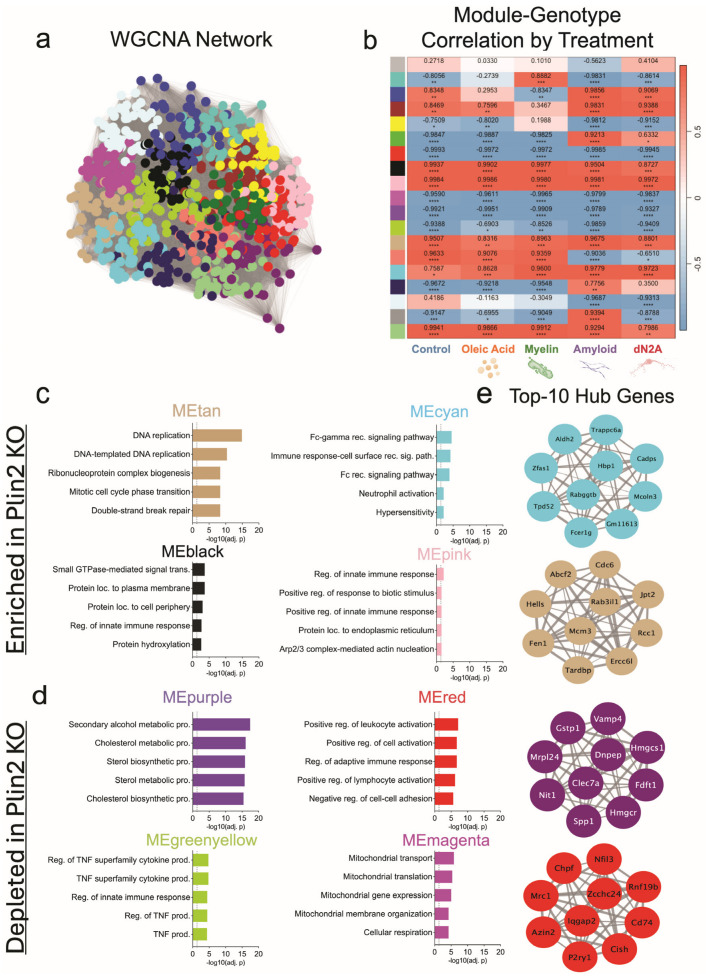
WGCNA of BV2 transcriptomes anchored to Plin2 KO. (**a**) WGCNA network built from variance-stabilized RNA-seq counts across all samples and treatments; nodes are genes, edges reflect topological overlap, and colors denote data-driven modules. (**b**) Heat map of module eigengene (ME) correlations with Plin2 KO within each treatment; cells show Pearson’s r with BH-adjusted *p*-values (red = positive, blue = negative). (**c**) KO-enriched modules (ME increased in KO); top Gene Ontology terms for representative modules. (**d**) KO-depleted modules (ME decreased in KO); top Gene Ontology terms for representative modules. (**e**) Top 10 hub genes for selected modules; networks display the ten genes with the highest intramodular connectivity (kME) per module, with edge thickness reflecting within-module connection strength. Stats: module–trait associations were computed as Pearson’s r between module eigengenes and genotype within each treatment with Benjamini–Hochberg FDR control. Hubs were defined as the top 10 genes by kME per module. Module GO enrichment results are in Appendix A. For all statistics, * *p* < 0.05; ** *p* < 0.01; *** *p* < 0.001; **** *p* < 0.0001.

**Figure 5 cells-14-01783-f005:**
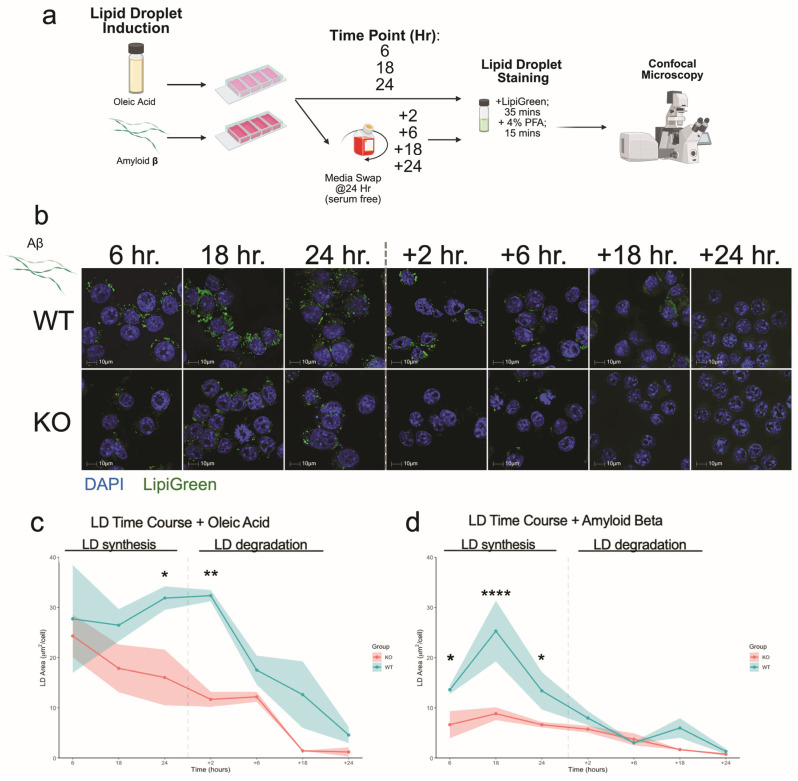
Plin2 loss alters lipid droplet accumulation and dynamics. (**a**) Experimental workflow. BV2 cells were exposed to oleic acid (OA) or amyloid beta (Aβ), fixed at 6, 18, and 24 h, or switched to serum-free medium for a chase and imaged at the indicated times after removal. Cells were stained with Lipi-Green (neutral lipids) and DAPI (nuclei) and imaged by confocal microscopy. (**b**) Representative fields for WT and Plin2 KO under Aβ at 6, 18, 24, +2, +6, +18, and +24 h. (**c**) OA time course. Total lipid droplet area per nucleus over time for WT and KO; WT shows a larger rise by 24 h and partial persistence after serum removal, whereas KO remains lower and trends toward baseline. Shading indicates mean ± SEM. (**d**) Aβ time course. WT displays an early rise peaking by 18 h and remaining above KO at 24 h; KO shows a minimal increase over the same window. Both genotypes return toward baseline by 48 h. Stats (**c**,**d**): two-way ANOVA within treatment (Genotype × Time) with Šídák adjusted post hoc tests; two-sided α = 0.05; data are mean ± SEM with *n* = 3. Exact adjusted *p*-values and full model outputs are provided in Appendix A. For all statistics, * *p* < 0.05; ** *p* < 0.01; **** *p* < 0.0001.

**Figure 6 cells-14-01783-f006:**
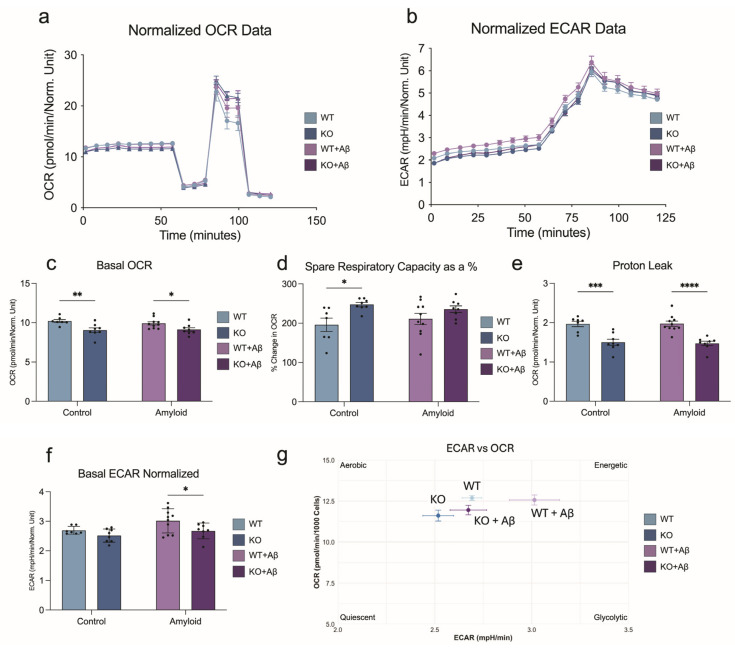
Plin2 deletion reshapes microglial bioenergetics. (**a**) Oxygen consumption rate (OCR) traces for WT and Plin2 KO BV2 cells under control or amyloid beta (Aβ) conditions, with sequential compound injections. (**b**) Extracellular acidification rate (ECAR) traces for the same groups. (**c**) Basal OCR. Plin2 KO operates at a lower basal respiration than WT under both vehicle and Aβ. (**d**) Spare respiratory capacity expressed as a percentage, calculated from FCCP-stimulated OCR relative to basal and maximal respiration. KO shows a larger reserve relative to WT. (**e**) Proton leak OCR following oligomycin. KO exhibits reduced leak compared with WT. (**f**) Basal ECAR. Under Aβ, KO relies less on glycolysis than WT. (**g**) Energetic phenotype plot mapping basal ECAR against basal OCR for each group to summarize shifts in oxidative vs. glycolytic profiles. Stats (**c**–**g**): two-way ANOVA (Genotype, Aβ) with Šídák post hoc tests; two-sided α = 0.05. Values were normalized to nuclei counts before analysis. Bars show mean ± SEM. Source data and full summary statistics are provided in Appendix A. For all statistics, * *p* < 0.05; ** *p* < 0.01; *** *p* < 0.001; **** *p* < 0.0001.

**Figure 7 cells-14-01783-f007:**
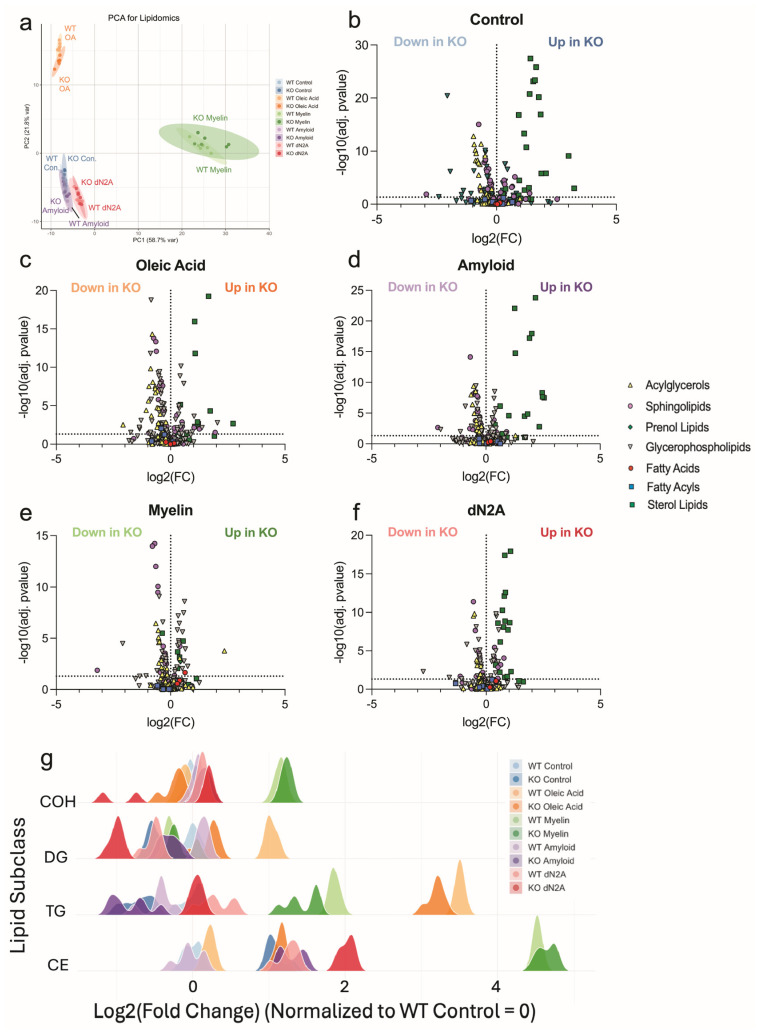
Targeted lipidomics reveals Plin2-dependent routing of neutral lipids. (**a**) Principal component analysis of lipid profiles from WT and Plin2 KO BV2 cells under control, oleic acid (OA), amyloid beta (Aβ), myelin, and dN2A treatments. (**b**–**f**) Volcano plots by condition showing log2 fold change (KO vs. WT) on the x-axis and −log10(adjusted *p* value) on the y-axis. Each point is a lipid species, with shape indicating lipid class (acylglycerols, sphingolipids, glycerophospholipids, fatty acids, prenol lipids, and sterol lipids). Labels indicate species increased in WT or increased in KO. Across Control, OA, Aβ, and dN2A, KO cells were depleted of triacylglycerols and diacylglycerols and enriched for cholesteryl ester. (**g**) Class-level distributions of log2 fold change (KO vs. WT) for representative lipid classes across conditions. Ridges summarize the pattern that WT favors TAG/DAG, whereas KO favors CE. Stats: species—per-feature linear models (KO−WT within each treatment) with Benjamini–Hochberg FDR (primary FDR < 0.05); subclasses—two-way ANOVA (Genotype, Treatment, Genotype × Treatment) with BH FDR across subclasses per term, plus pairwise KO−WT contrasts BH-adjusted across subclasses within each treatment. Exact adjusted *p*-values and full model outputs are provided in Appendix A.

## Data Availability

RNA-seq count matrices/DEGs, WGCNA module memberships/eigengenes, Seahorse source metrics, and lipidomics raw/processed tables are provided in Appendix A. Additional analysis pipelines and scripts used in this study can be provided by the corresponding author upon reasonable request.

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
