# Peer review of "Knockout of Perilipin-2 in Microglia Alters Lipid Droplet Accumulation and Response to Alzheimer’s Disease Stimuli"

_cells, 2025, doi:10.3390/cells14221783_

Round 1
Reviewer 1 Report
Comments and Suggestions for Authors
In this manuscript, Stephens and Johnson demonstrate that Plin2, a lipid-droplet associated protein, is an important regulator of lipid droplet size and accumulation in microglia. Plin2 expression increases with aging and in Alzheimer’s disease and is prominently detected in lipid-droplet accumulating microglia. This study provides strong evidence that loss of Plin2 decreases lipid droplet burden and improves the phagocytic and metabolic profile of microglia in response to various disease-relevant stimuli, including amyloid-b. Since lipid accumulation in microglia is associated with aging and neurodegeneration and is linked to impaired phagocytosis and chronic inflammation, targeting Plin2 may serve as a therapeutic strategy to modulate disease-associated microglia and neuroinflammation.
Overall, the study is well designed and would be of great interest to the field. These findings will expand our understanding of the etiology of Alzheimer’s disease. While the study is compelling, a few points need to be addressed to make the manuscript suitable for publication. Below is a list of major and minor comments that I hope the authors will find helpful.
Major comments:
- Immunofluorescent staining of mouse brain slices is missing from the methods section.
- The authors also need to include a description about how the images were quantified in the methods. For example, how were the lipid droplets detected/thresholded and counted?
- In figure 1, the quantification shows less LD area and average size, but no difference in the number of LDs. But the image selected shows a lot less lipid droplets roughly the same size. Is this an issue with how the images were thresholded? If not, the authors should replace the representative image to better reflect their quantification.
- The PLIN2 staining in figure 1D looks cytosolic. Shouldn’t this be more closely associated with LDs?
- Under basal/control conditions, there is no significant difference in LD number, size or area between WT VS Plin2 KO microglia, but there is a difference in the phagocytosis of zymosan particles under these conditions (Figure 2B). Do the authors know what this effect could be attributed to, since the WT microglia under these control conditions would not be considered ‘lipid-laden’? Could Plin2 regulate this pathways independent of lipid storage?
- To generate apoptotic debris, how long were the cells incubated in their flasks with the lid tightened? What percentage of the cells are dead? Also, for the conditioned media collected from the N2A cells, did it contain more than just apoptotic N2A cells? Is this controlled for in the assays?
- For Figure 5B, it’s difficult to see the LipGreen-positive lipid droplets in the images. Can the authors provide different representative images or a magnified panel?
- Along the same lines, for Figure 5, can the authors provide representative lipid droplet images of WT and Plin2 KO cells under OA conditions?
- For the ‘acylglycerol’ category of the volcano plots in Figure 7b, what is the proportion of TAGs VS DAGs? Although there is a clear difference in acylglycerols in WT VS KO cells under control conditions, the authors did not find any significant differences in lipid droplet size, area or number under these same conditions. Does that mean that under control conditions, there is a switch in lipid droplet composition, where WT cells primarily contain TG-rich lipid droplets while Plin2 KO cells contain CE-rich lipid droplets?
- The authors could look at lipid droplet accumulation with and without DGAT inhibitors to further confirm that Plin2 promotes TG-rich lipid storage in microglia.
- Figure 3 shows that KO cells downregulate sterol and broader lipid-biosynthetic pathways. Does that mean that the enrichment in CE levels in KO cells shown in Figure 7 is a homeostatic response to this downregulation? If the CE is not be used for lipid storage, can the authors speculate what this pool of CE is doing in the cells?
Minor comments:
- Western blot protocol used for the knockout validation is missing from the methods section.
- For the image panels in the paper, it would be nice if the there was some white space between the different images.
- For Line 72 should ‘an’ be ‘a’?
- For Figure 2A and 5B, the images are missing the dye labels.
- Figures 6A and 6B are not referenced in the main text.
- For Figure 6C-F, it would be nice to include the color-coded legend for each panel to quickly differentiate treatments.
- In Line 476 should ‘stabilizes’ be ‘stabilizing’?
- Since Ab has the most robust effect on WT VS KO cells, the authors could test how Ab affects zymosan uptake like they did in Figure 2 for OA conditions.
Reviewer 2 Report
Comments and Suggestions for Authors
In the CNS LDs accumulate in response to stress, inflammation and insults (myelin, Abeta) reprogramming their state to LD-accumulating microglia or LDAMs. This impacts phagocytosis and pro-inflammatory profiles, but how these pathways are mechanistically connected remained unanswered. Stephens and Johnson have explored this by knocking out perilipin2, which has a central role in LD formation, in a microglial cell line.
KO of plin2 resulted in smaller sized LDs concomitant with increased phagocytic capacity. The authors performed several omics analysis, including bulk RNAseq and lipid profiling, combined with assessing mitochondrial energetics, that further supported a role for plin2 as a key regulator in LD formaton, metabolic and immune responses to different stressors. This is a limited study confined to a microglial cell line but overall the experimental quality is high and the data support the overall conclusions. I have only a few comments that the authors should address.
The Result section starts with IHC of 5xFAD brain, showing that LDs accumulate in microglia around plaques. However the M&M mention a APOE4-TR;5xFAD model. (1) What model is used and (2) why are the two mouse genotypes not compared? Would one expect to see an aggravation of the LD phenotype? If these comparison is not made, I wonder what the relevance is of including the data on the 5xFAD mice (a pln2KO; 5xFAD would be even more relevant): for the remainder of the manuscript, only BV2 cells are used.
Overall, the study is limited to BV2 which diminishes the translational relevance. If the authors should see a possibility to validate some of their major findings in more appropriate and disease relevant human models, this would strongly increase the relevance of the study.
Minor comment:
-line 310 states ‘under AD-relevant stimuli such as myelin and apoptotic neurons’. I would consider myelin rather relevant in the context of MS, not AD. Also strange that the next sentence starts with ‘By contrast, Abeta exposure elicited…’ which is a true AD stressor. Please clarify and re-formulate.
Reviewer 3 Report
Comments and Suggestions for Authors
The manuscript by Stephens and Johnson describe changes induced by perilipin 2 (Plin2) KO in immortilized microglia cell line. The data suggest a central role of Plin2 in shaping lipid droplet biology, with impacts in transcription, metabolic and functional parametrs of BV2 microglia.
the authors show that Plin2 deletion significantly reduces lipid droplet accumulation under basal and oleic acid treatment. Functionally, Plin2 KO cells also exhibit enhanced zymosan phagocytic activity, even under OA treatment.
Regardinf their metabolic signature, Aβ1-42 incubation induced strong changes in genes related to mitochondrial function, which was later further characterized by seahorse experiments, that showed reduced glycolysis, preserved mitochondrial respiration, and greater spare respiratory capacity in Plin2 KO cells.
The evidence presented support the role of Plin2 in LD metabolism and microglia function, but a few points must be addressed before the manuscript can be accepted for publication:
Images resolution has to be addressed. Very poor quality
Line 431 - cholesterol esters (CE)
Clarification of Animal Model
Please clarify whether the animal model used in Figure 1 corresponds to the 5xFAD line or the APOE4-TR;5xFAD cross, as described in the Materials and Methods section (2.2).
Functional Validation under Amyloid-β Treatment
The authors convincingly show that Plin2 deletion reduces lipid droplet (LD) accumulation and enhances zymosan phagocytosis following oleic acid (OA) treatment (Figures 1 and 2). However, it would be important to include comparable analyses under amyloid-β (Aβ1-42) exposure. Demonstrating LD accumulation and phagocytic activity in Aβ-treated microglia would strengthen the relevance of the findings to Alzheimer’s disease–related contexts.
Transcriptomic Divergence under OA Treatment (Figure 3b)
The RNA-seq results show that OA-treated BV2 cells exhibit a transcriptomic profile markedly distinct from those exposed to Aβ or dN2A-derived media. The authors should consider using the AD cocktail approach (OA + Aβ1-42, myelin+ dN2A media) in future experiments to better model the complex in vivo environment
Experimental Design for LD Turnover (Figure 5)
In Figure 5, authors describe LD dynamics (formation vs degradation). It would be most informative if authors could delete Plin2 in BV2 microglia previously treated with OA to induce LD formation, as well as, use the AD cocktail (or OA + Abeta combination) in these experiments.
Round 2
Reviewer 1 Report
Comments and Suggestions for Authors
The authors have addressed all of my comments. I support the publication of the study.
Author Response
Thank you for helping improve our manuscript. We’re grateful for your support to proceed with publication.
Reviewer 3 Report
Comments and Suggestions for Authors
Minor corrections
Line 413: ingest
Line 533: please define DGAT1 here
Author Response
We’ve fixed both points: corrected the wording at line 413 and added a definition of DGAT1 (diacylglycerol acyltransferase-1) at line 533. Thank you for catching these and helping improve our manuscript.